# Structure of *Leishmania donovani* 6-Phosphogluconate Dehydrogenase and Inhibition by Phosphine Gold(I) Complexes: A Potential Approach to Leishmaniasis Treatment

**DOI:** 10.3390/ijms24108615

**Published:** 2023-05-11

**Authors:** Isabell Berneburg, Michaela Stumpf, Ann-Sophie Velten, Stefan Rahlfs, Jude Przyborski, Katja Becker, Karin Fritz-Wolf

**Affiliations:** 1Biochemistry and Molecular Biology, Interdisciplinary Research Center, Justus Liebig University, 35392 Giessen, Germany; isabell.berneburg@ernaehrung.uni-giessen.de (I.B.);; 2Max Planck Institute for Medical Research, 69120 Heidelberg, Germany

**Keywords:** leishmaniasis, 6-phosphogluconate dehydrogenase, pentose phosphate pathway, auranofin, phosphine gold(I)-complexes

## Abstract

As unicellular parasites are highly dependent on NADPH as a source for reducing equivalents, the main NADPH-producing enzymes glucose 6-phosphate dehydrogenase (G6PD) and 6-phosphogluconate dehydrogenase (6PGD) of the pentose phosphate pathway are considered promising antitrypanosomatid drug targets. Here we present the biochemical characterization and crystal structure of *Leishmania donovani* 6PGD (*Ld*6PGD) in complex with NADP(H). Most interestingly, a previously unknown conformation of NADPH is visible in this structure. In addition, we identified auranofin and other gold(I)-containing compounds as efficient *Ld*6PGD inhibitors, although it has so far been assumed that trypanothione reductase is the sole target of auranofin in *Kinetoplastida*. Interestingly, 6PGD from *Plasmodium falciparum* is also inhibited at lower micromolar concentrations, whereas human 6PGD is not. Mode-of-inhibition studies indicate that auranofin competes with 6PG for its binding site followed by a rapid irreversible inhibition. By analogy with other enzymes, this suggests that the gold moiety is responsible for the observed inhibition. Taken together, we identified gold(I)-containing compounds as an interesting class of inhibitors against 6PGDs from *Leishmania* and possibly from other protozoan parasites. Together with the three-dimensional crystal structure, this provides a valid basis for further drug discovery approaches.

## 1. Introduction

Leishmaniasis is a vector-borne infectious disease that is classified by the World Health Organization as one of 20 poverty-related and neglected tropical diseases. It is caused by the protozoan parasite of the genus *Leishmania*, which belongs to the *Trypanosomatidae* family. Every year, 0.7 to one million people become infected, and 20,000 to 30,000 people die from leishmaniasis in tropical, subtropical and southern European regions. The species *Leishmania donovani* causes the most severe form, also known as kala-azar or “black fever” and is often fatal if untreated [1,2]. Currently, there are only a few drugs available for leishmaniasis treatment. Most of them have severe side effects and show highly divergent activities across subspecies. After decades of use, resistant parasites have emerged [3,4,5], which is why ambitious efforts to develop new drug targets and antileishmanial agents are urgently required. 

Throughout their life cycle, many parasites, and *Leishmania* parasites in particular, are exposed to high levels of oxidative stress. For example, the tissue stages of *Leishmania*—amastigotes—reside in mammalian macrophages and must withstand the action of reactive oxygen and nitrogen species (ROS/RNS) produced by enzymes such as NADPH oxidase or nitric oxide (NO) synthase [6,7,8,9]. The antioxidant defense and redox balance of the parasites rely on the trypanothione-dependent thiol system, including trypanothione and trypanothione reductase, and depend on NADPH as the primary electron source [10,11]. As the pentose phosphate pathway (PPP) is the main NADPH supplier in most parasites, the enzymes of this key metabolic pathway are considered promising drug targets for the control of unicellular parasites such as *Leishmania*, *Trypanosoma* [3,12,13] and *Plasmodium* [14,15]. NADPH is produced in the first, unidirectional oxidative branch of the PPP. The second, non-oxidative branch of the PPP ends with ribose 5-phosphate, a crucial product for nucleotide biosynthesis. In total, two molecules NADPH are produced by the oxidative PPP. Glucose 6-phosphate dehydrogenase (G6PD) (EC 1.1.1.49) is the first enzyme of this pathway and catalyzes the oxidation of glucose 6-phosphate (G6P) from glycolysis to 6-phospho-D-glucono-1,5-lactone, whereby the cofactor NADP^+^ is reduced to one molecule, NADPH. 6-phosphocluconate dehydrogenase (6PGD) (EC 1.1.1.44) is the third enzyme of the oxidative branch and generates a second NADPH molecule by decarboxylating 6-phosphogluconate (6PG) to ribulose 5-phosphate (R5P) and CO_2_ [11]. In trypanosomatids, enzymes of the PPP as well as other glycolytic enzymes are localized in the cytosol as well as in peroxisome-related glycosomes, probably allowing a fast adaption to environmental changes and preventing accumulation of toxic metabolites [16,17,18]. G6PD, the first enzyme of the PPP, is considered to be the rate-limiting enzyme of this pathway. However, 6PGD activity also influences the PPP and glycolytic enzymes. 6PGD inhibition leads to an accumulation of toxic 6PG, which in turn inhibits the glycolytic enzyme glucose 6-phosphate isomerase [19]. This triggers a positive feedback loop leading to more glucose 6-phosphate flux via the PPP and consequently inhibition of glycolysis [3,20,21] and has been shown to be lethal to some eukaryotes, such as *Trypanosoma brucei* [20], which belong to the same family as *Leishmania*. However, RNAi knockdown of 6PGD in bloodstream *Trypanosoma brucei* (*Tb*6PGD^RNAi^), which leads to reduced 6PGD activity and subsequent cell death, suggests that loss of the oxidative branch of the PPP is lethal, rather than inhibition of glycolysis or lack of ribose 5-phosphate for nucleotide biosynthesis, as neither the addition of fructose nor ribose prevented cell death of the *Tb*6PGD^RNAi^ cell line [22]. The suitability of *Tb*6PGD as drug target is further supported by *Tb*6PGD inhibitors that enable selective inhibition compared to the mammalian counterpart [23]. Since *Leishmania* 6PGD shares high sequence similarity to other trypanosomatid 6PGDs and at the same time differs significantly from its human counterpart, 6PGD has also been proposed as a drug target in *Leishmania* [24]. Moreover, strains resistant to commonly used antileishmanial drugs such as miltefosine [25], amphotericin-B [26] and sodium antimony gluconate [27], whose mode of action is thought to be associated with increased ROS production, showed an upregulation of the PPP [9]. 

Recently, the FDA-approved drug auranofin, which has been used for decades against rheumatoid arthritis [28,29], received increasing attention because of its potential to be repurposed as an antiparasitic drug [30,31,32,33,34]. Its antiparasitic activity has also been demonstrated for *Trypanosoma* [35,36] and *Leishmania* spp. [37,38,39]. Although the mechanism of action (MOA) of auranofin and other gold(I)-containing drugs is still a matter of debate as they appear to have multiple targets, the suggested MOA in *Kinetoplastida* is the inhibition of the redox enzyme trypanothione reductase (TR) [36,37,39], and thus comparable to the target thioredoxin reductase (TrxR) in mammals [40,41,42] and several other parasites [31,43]. Inhibition of these redox enzymes is associated with a disruption of redox homeostasis, leading to severe oxidative stress and cytotoxic effects in vitro [38,39,44]. However, recent studies on auranofin resistance in several protozoan parasites, such as *Toxoplasma gondii*, *Entamoeba histolytica* and *Giardia lamblia*, suggest that TrxR may not be the sole target of auranofin, as auranofin-resistant parasite strains accumulate less ROS, without any mutations in the TrxR gene or changes in TrxR expression levels [34,45,46]. These studies prompted us to test auranofin against other redox enzymes, such as the NADPH-producing enzymes of the PPP. Indeed, auranofin and other gold(I)-containing compounds inhibit *Ld*6PGD in the low micromolar range, whereas the human enzyme is hardly affected at these concentrations. This gives reason to reinterpret previous in vivo studies with auranofin in *Kinetoplastida*, as its MOA can no longer be attributed solely to TR inhibition. Mode-of-inhibition studies of auranofin indicated an initial competition with the 6PG binding site followed by a rapid irreversible inhibition. In addition, we solved the three-dimensional crystal structure of *Ld*6PGD in complex with NADP(H), revealing a previously unknown conformation of NADPH. Finally, these structural and biochemical insights into *Ld*6PGD provide an excellent basis for further structure-based studies on gold(I)-containing compounds as an interesting class of protozoan 6PGD inhibitors. 

## 2. Results and Discussion

### 2.1. Production, Oligomerization Behavior and Kinetic Characterization of Ld6PGD wt

*Leishmania donovani* 6PGD was recombinantly produced using a pET28a vector in *E. coli* (DE3) cells and resulted in a final yield of 1–4 mg soluble, pure and active *Ld*6PGD per liter of *E. coli* culture (Appendix A). 

To investigate the oligomerization behavior of *Ld*6PGD, we performed size exclusion chromatography (SEC) under different conditions. Under native conditions, the SEC profiles of *Ld*6PGD wt revealed one stable peak representing a molecular mass of 94–111 kDa (Figure 1a), which is equivalent to the dimeric form of the *Ld*6PGD wt and in accordance with previously published SEC profiles from *Leishmania* 6PGDs [24,47]. Varying *Ld*6PGD concentrations did not affect the elution profile (Appendix A). Moreover, the elution pattern of *Ld*6PGD was neither affected by reductive (5 mM DTT) nor by oxidative conditions (2 mM H_2_O_2_) (Figure 1a), suggesting the formation of the dimer is independent from intermolecular disulfide bonds. Most other 6PGDs are active as homodimers and also exhibited a dimer in the crystal structure, such as human 6PGD (*Hs*6PGD) [48,49,50] (PDBIDs: 5UQ9 [50]; 2JKV, to be published) or 6PGDs from *Ovis aries* (PDBID: 1PGO [51]), *Saccharomyces cerevisiae* (PDBID: 2P4Q [52]), *Plasmodium falciparum* (*Pf*6PGD) (PDBID: 6FQX [49]), *Lactococcus lactis* (PDBID: 2IYO [53]), *Geobacillus stearothermophilus* (PDBID: 2W8Z [54]) or from the related *Trypanosoma brucei* 6PGD (*Tb*6PGD) (PDBID: 1PGJ [55]). Hanau et al. [21,56] found that *Tb*6PGD exists in a dynamic dimer–tetramer equilibrium, which is shifted towards the tetramer in presence of the product NADPH. Therefore, we tested whether the oligomerization of *Ld*6PGD is similarly affected and performed SEC in the presence of substrates and products. In contrast to the *Tb*6PGD [56], our studies do not support a dimer–tetramer equilibrium for *Leishmania* 6PGDs, since the dimer remained unaffected under the conditions tested (Figure 1b).

Steady-state kinetics revealed a specific activity of the *Ld*6PGD wt dimer of 33.3 ± 5.1 U·mg^−1^ with apparent *K*_M_ values of 20.2 ± 2.9 µM for the substrate 6PG and 13.7 ± 1.2 µM for the cosubstrate NADP^+^. Based on these values, we calculated a catalytic efficiency of 1.5 ± 0.02 µM^−1^·s^−1^ for 6PG and of 2.2 ± 0.2 µM^−1^·s^−1^ for NADP^+^ (Table 1). These values are very similar to other 6PGDs characterized so far [47,49,57,58] and nearly identical to 6PGDs of the related *Trypanosoma* species *T. cruzi* and *T. brucei* [59,60], with which *Leishmania* 6PGD shares a high sequence identity of approximately 70%. 

### 2.2. Crystallization and Structure Determination of Ld6PGD

We obtained orthorhombic (P2_1_2_1_2_1_) crystals of the *Ld*6PGD wt in complex with NADP(H) (PDBID: 8C79). The structure was solved at 3.1 Å by the molecular replacement method, using *T. brucei* 6PGD (PDBID: 1PGJ) as a template, which shares 72% sequence identity with *Ld*6PGD (Figure 2). The crystal contained two monomers in the asymmetric unit (AU), which were essentially similar to an RMSD of 0.4 Å (478 residues). The omit maps displayed clear density for one phosphate ion in each subunit and one NADP(H) molecule in subunit A. In subunit B, most of the NADP(H) molecule is well defined by electron density, except for the nicotinamide and the adenine moieties, which are not visible. The structure contains a complete model of *Ld*6PGD, comprising residues 1–478, two NADP(H) molecules, two phosphate ions and one water molecule. Despite the relatively high Wilson B-factor of 129 Å^2^, the structure is well defined by the electron density and displays good stereochemistry. All data collection and refinement statistics are summarized in Table 2. 

### 2.3. Overall Structure of Ld6PGD

The primary sequences as well as the three-dimensional structures within the 6PGD family are highly conserved. This is also true for *Ld*6PGD, which shares high sequence similarity and identity with other 6PGDs of this family (Figure 2). Similar to other members of the 6PGD family, *Ld*6PGD is a functional homodimer and adopts the canonical 6PGD fold (Figure 3) with RMSD values for the dimer of 1.4 Å (956 residues), 1.0 Å (954 residues) and 1.8 Å (878 residues) to human (37% sequence identity; PDBID: 2JKV; NADP^+^-complexed structure), *T. brucei* (72% sequence identity; PDBID: 1PGJ; apo structure) and *P. falciparum* 6PGD (34% sequence identity; PDBIDs: 6FQZ; 6PG-complexed structure), respectively. 

The *Ld*6PGD monomer contains the typical 6PGD domains: the smaller N-terminal “Rossmann-like” domain 1 (S1-K177), which is composed of a mixed parallel and anti-parallel seven-stranded β-sheet, and the larger, mainly α-helical domain 2 (A180-Q478). The C-terminal tail of domain 2 (M438-Q478) invades the neighboring subunit so that each active site is built by residues of both subunits (Figure 3). Each subunit of the homodimer comprises a NADP^+^ and a 6PG binding site (Figure 4 and Appendix A). The 6PG binding site is located in a cleft of domain 2 and includes the following residues: S130-G132, K185, N189, E192, Y193, K262, T264, and R289 from one subunit and R453′ and H459′ from the adjacent monomer. NADP(H) is bound by conserved residues of domain 1. Upon catalysis, the binding site of the nicotinamide ring and 6PG overlaps and is formed by M12, E192, N104, I129-G132, K185, N189 and F456′ and H459′ from the other subunit.

### 2.4. Active Site in the Ld6PGD Structure

In our *Ld*6PGD structure, NADP(H) is located at the conserved NADP^+^ binding site (Figure 4a). The amino acids N31, R32 and T33 interact with one of the phosphates, N31 and N104 with the ribose moieties and the diphosphates are bound by main chain atoms of residues V11 and M12. Via van der Waals forces, I129 and S130 interact with the nicotinamide ring, which is also hydrogen bonded to a phosphate ion. This phosphate ion is located at the same position as the phosphate moiety of homologous 6PG-complexed structures (PDBIDs: 2IYO, 6FQZ) (Figure 4b) and also interacts with homologous strictly conserved residues (Y193, K262, T264, R289 and R453′, H459′ from the adjacent subunit) in a similar manner to other 6PGDs. Moreover, a hydrogen bond (2.6 Å) is visible between the phosphate ion and the nicotinamide ring (N7N atom) of NADP(H). It is known for many 6PGDs that a flexible active site loop adopts an open and a closed conformation depending on substrate binding [49]. The active site loop adopts the closed conformation when the 6PG binding site is occupied by ligands or at least by anions such as phosphate, sulfate or citrate, all of which are capable of interacting with residues of the C-terminal tail. According to this, the active site loop residues (D257-T262) of the *Ld*6PGD structure, complexed with a phosphate at the 6PG binding site, adopt the closed conformation.

### 2.5. Mechanistic Considerations

There is clear electron density for an NADP(H)-molecule and a phosphate ion in the active site, which are connected by a hydrogen bond (Figure 4 and Appendix A). Since *Ld*6PGD was crystallized under catalysis, in the presence of both substrates, 6PG and NADP^+^, it could be also the products (R5P, NADPH) or an intermediate state of substrate and product that are visible in the active site. A structural comparison of the *Ld*6PGD structure with homologous NADP^+^-complexed structures (*Ovis aries*, PDBID: 1PGN; *Lactocuccus lactis*, PDBID: 2IYP; *Hs*6PGD, PDBID: 2JKV) revealed that the NADP(H) molecule is bound very similarly in these structures, except for the nicotinamide ring (Figure 5a), which adopts the same conformation as in the NADPH-complexed 6PGD structure from *Ovis aries* (PDBID: 1PGO) (Figure 5b). 

It has been shown earlier that the orientation of the nicotinamide ring indicates whether NADP^+^ or NADPH is bound [51]. When the cosubstrate is reduced to NADPH, the nicotinamide ring rotates from syn to anti and points therefore towards the 6PG binding site, which also involves a rotation of the ribose and phosphate moieties so that the ribose oxygen atoms then point towards the solvent (Figure 5b). Considering only the position of the nicotinamide ring, which is similarly oriented as in the NADPH-complexed structure from *Ovis aries* (PDBID: 1PGO), we conclude that NADPH is bound in the *Ld*6PGD structure. On the other hand, the position of the ribose unit and phosphates in the *Ld*6PGD structure corresponds to the NADP^+^-complexed structures, thus NADP^+^ would be bound in the *Ld*6PGD structure (Figure 5a).

It is known that the product NADPH acts as a competitive inhibitor against NADP^+^ and as a non-competitive inhibitor against 6PG [61,62,63]. Since crystallization with an excess of 6PG and NADP^+^ in the crystallization buffer, and thus under catalysis, could have resulted in a feedback inhibition by high concentrations of the product NADPH, we suggest that *Ld*6PGD is complexed with NADPH in its inhibitory conformation, in which the nicotinamide ring occupies part of the 6PG binding site. Consequently, we suggest that the bound phosphate moiety is from 6PG and not from the product R5P. As the nicotinamide ring occupies part of the 6PG binding pocket, we hypothesize that only the phosphate component of 6PG can bind at the usual site, while the rest of the 6PG molecule must be flexible and therefore not visible in the electron density (Figure 4a and Figure 5b). Theoretically, the postulated NADPH conformation in the *Ld*6PGD structure could also be an artefact due to the phosphate ion forcing the positively charged nicotinamide ring of NADP^+^ into this position, and thereby forming a hydrogen bond between these two entities. In this case, this interaction should also be seen in the human 6PGD structure (PDBID: 2JKV) complexed with NADP^+^ and a sulfate ion. However, this is not the case, the cofactor adopts the NADP^+^ typical conformation, supporting our hypothesis that in our structure, NADPH is bound in its inhibitory conformation.

### 2.6. Ld6PGD Is a Target of the Gold Inhibitor Auranofin 

In this study, we tested auranofin (Figure 6), a gold(I)-containing drug that has been used for decades to treat rheumatoid arthritis [28,29], against *Ld*6PGD. We determined IC_50_ values of 8.6 ± 1.0 µM (K*_i_* = 3.8 ± 0.4 µM) against recombinant *Ld*6PGD (Table 3). Interestingly, the plasmodial *Pf*6PGD was inhibited at similar concentrations with IC_50_ values of 20.1 ± 1.5 µM, but the human *Hs*6PGD was not inhibited up to a concentration of 100 µM auranofin (Table 3). Although several studies have shown that auranofin is also active against *Kinetoplastida* parasites [36,37,39], to our knowledge, no study has yet identified 6PGD as a potential target of auranofin in these parasites. This raises the question of whether the inhibition of *Ld*6PGD is relevant or negligible for the MOA of auranofin in *Leishmania* parasites. Depending on the *Leishmania* species and parasite stage in vitro, EC_50_ values of auranofin and other gold(I)-containing derivatives reported in the literature range from low micromolar (10^−5^ to 10^−6^) [37,64] to nanomolar (10^−7^ to 10^8^) [38,44] concentrations, making gold(I)-containing drugs similarly effective to common antileishmanial drugs such as antimony in vitro [65,66]. Based on co-crystallization and inhibition studies, it was previously suggested that this high in vitro efficacy is solely due to the inhibition of trypanosomatid TR [37,64]. The authors reported *K_i_* values of 155 ± 35 nM against recombinant TR from *Leishmania infantum* (*Li*TR) [37]. However, although these studies suggest that auranofin inhibits TR more efficiently than *Ld*6PGD, the inhibition of *Ld*6PGD in the lower micromolar range cannot be ignored, especially when interpreting the MOA of auranofin in vitro and in vivo. The cytotoxic effect of auranofin on trypanosomatid parasites has previously been explained by an increase in ROS and induced apoptotic-like cell death and is consistent with the hypothesized MOA of TR inhibition [38,39,44]. However, these effects were only observed at concentrations of 10–50 µM, which is surprising, given the nanomolar activity against *Li*TR. Since, in our studies, *Ld*6PGD was completely inhibited at 50 µM auranofin, the MOA could also be explained by the inhibition of the *Ld*6PGD. Some in vitro studies reported EC_50_ values significantly lower than the *K*_i_ of auranofin against *Li*TR [38,44]. This could be explained by the fact that inhibition of TR as a central redox enzyme could also affect the activity of other downstream redox enzymes and consequently the overall redox balance. However, our observations also suggest a synergistic effect based on dual TR and 6PGD inhibition, as the activity of TR as a NADPH consumer is directly dependent on the activity of 6PGD and G6PD as NADPH producers, which would accelerate the generation of ROS and consequently the cytotoxic effect of auranofin on trypanosomatid parasites. Finally, our data suggest that the MOA of auranofin should be reconsidered and the contribution of 6PGD inhibition to the overall MOA of auranofin in *Kinetoplastida* further investigated.

### 2.7. Auranofin Competes with the 6PG Binding Site and Binds Irreversibly to Ld6PGD

To investigate the mode of inhibition (MOI) of auranofin against *Ld*6PGD, various compound concentrations were titrated against the substrates NADP^+^ and 6PG, and the relationship between initial velocity and substrate concentrations was analyzed. Michaelis–Menten kinetics indicate a non-competitive inhibition against NADP^+^, as the *K*_M_ remained stable and V_max_ decreased with increasing auranofin concentrations (Figure 7a). In contrast, the MOI against 6PG cannot be clearly classified. Both V_max_ and *K*_M_ increased with increasing concentrations of auranofin, corresponding to a mixed type of inhibition. However, the strong increase in *K*_M_ of 6PG suggests competitive inhibition (Figure 7b). This was confirmed by determination of IC_50_ values at saturation. Incubation of the enzyme with auranofin and 6PG at saturation resulted in IC_50_ values of 40.5 ± 1.9 µM. When instead auranofin and the enzyme were incubated with NADP^+^ at saturation, the IC_50_ was unaffected, again supporting the non-competitive inhibition against NADP^+^.

In addition, we investigated whether the inhibition of *Ld*6PGD by auranofin is reversible or irreversible. For this purpose, the enzyme was first incubated with a high concentration of auranofin (pre-dilution, 30 µM), followed by a dilution below the IC_50_ (3 µM, post-dilution) and incubated again to allow compound dissociation (Figure 8). Compared to controls containing either no auranofin (CTL, 0 µM) or an auranofin concentration equal to the post-dilution concentration (CTL, 3 µM), *Ld*6PGD activity could not be restored after dilution, indicating an irreversible type of inhibition. This is consistent with previous studies of auranofin and other gold(I)-containing compounds, which also observed irreversible binding of this class of compounds to their targets [41,67,68,69]. The irreversible inhibition of *Ld*6PGD also explains the ambiguous competitive inhibition mode against 6PG described above. We therefore suggest that the competition of auranofin with 6PG is followed by a rapid irreversible reaction in which a covalent product is formed, explaining why high concentrations of 6PG increase the IC_50_ but do not abolish the inhibition. Several studies indicated that the true pharmacophore of auranofin is triethylphosphanuidylgold(1+) [Au(PEt)_3_]^+^ or simply the gold ion Au(I) alone, rather than the whole auranofin molecule which binds to target proteins, such as TrxR and TR [31,37,41,42,70] or metalloenzymes such as ureases and metallo-β-lactamases [69,71]. In vivo, the thiosugar of auranofin acts more as a carrier ligand and is already displaced by protein thiols or free thiols in the blood. In vitro, selenocysteine or cysteine residues, typically activated by histidines or metal-ions, are necessary to release [Au(PEt)_3_]^+^ or Au(I) from the thiosugar or other ligands and enable covalent and thus irreversible binding to the target proteins [37,67,68,69,70,71,72]. Since the active site of *Ld*6PGD lacks a typical cysteine pair, we investigated whether only the gold moiety is responsible for *Ld*6PGD inhibition or whether the thiosugar or even the whole auranofin molecule is involved in the MOI. For that purpose, we tested two gold(I)-containing compounds—GoPi-sugar, sharing the thiosugar group with auranofin, and Ph_3_PAuCl (chloro(triphenylphosphine)gold(I)), an auranofin variant which in contrast lacks the thiosugar group (Figure 6). With 9.4 ± 2.1 µM, the IC_50_ of GoPi-sugar against *Ld*6PGD is almost identical to that of auranofin. Interestingly, Ph_3_PAuCl showed even better activity against *Ld*6PGD, with an IC_50_ of 2.1 ± 0.3 µM (Table 3). We suggest that it is again the gold moiety ([Au(PEt)_3_]^+^ or Au(I)) that is mainly responsible for the MOI, as it is the only common feature of the three compounds tested. The higher IC_50_ of auranofin and GoPi-sugar compared to Ph_3_PAuCl may be due to the large thiosugar, hindering the access to the target site. 

It remains to be determined to which site of the enzyme the gold moiety binds. As described above, Au(I) or [Au(PEt)_3_]^+^ of various gold(I)-compounds bind preferentially to cysteines or selenocysteines within the active site of their targets. Therefore, we analyzed the nine cysteines of *Ld*6PGD, three of which are *Leishmania*-specific (C241, C294 and C410) (Figure 2). Potential cysteine pairs not found in the human 6PGD are C183-C347, C294-C347′ and C336-C410. However, in the *Ld*6PGD structure, no disulfide bond is visible between the respective cysteines, and there are no neighboring residues such as histidine and aspartic acid, or alternatively glutamic acid, which could activate the cysteines to covalently bind the gold moiety. In addition, the corresponding cysteines are not in close proximity to the active site, which is why the competitive binding mechanism towards 6PG could only be explained by long chain forces, but this is not evident from the structure. Therefore, based on our kinetic studies, we suggest that the gold moiety binds directly to the 6PG binding site, but by a different mechanism than that previously described. The gold moiety could also be coordinated between M12 and H459 within the 6PG binding site. Asp454 and Glu133 could accept a proton from M12 and H459, providing the necessary negative partial charge to allow the covalent binding to the gold moiety (Figure 9). However, these residues are conserved among 6PGDs, which raises the question of why *Ld*6PGD and *Pf*6PGD are efficiently inhibited (Table 3), but the human 6PGD is not inhibited by the gold(I)-compounds we tested. In contrast to protozoan and prokaryotic 6PGDs, 6PGDs from multicellular species, such as from *Homo sapiens*, *Ovis aries*, *Saccharomyces cerevisiae* or from *Schistosoma* parasites, have an elongated C-terminal domain of about 10–15 residues that could cover the active site and thus sterically prevent auranofin from binding to the 6PG active site. However, while residues S478, S479 and Y481 of *Hs*6PGD (PDBID: 2JKV) are involved in NADP^+^ binding (Figure 9), the exact function of this C-terminal elongation or whether it alters its conformation in a substrate-dependent manner is not fully understood. For example, it is fully visible in the NADP^+^-complexed *Hs*6PGD structure and covering the active site (PDBID: 2JKV), but it is not visible in all other solved 6PGD structures from multicellular species, whether the substrate and cosubstrate are bound or not (PDBIDs: 4GWG, 4GWK, 1PGJ, 1PGO, 1PGN, 1PGQ, 2P4Q). In addition, there are conflicting studies as to whether the substrates bind by an ordered or random sequential binding mechanism [49,52,59,61,73,74], which could influence the position of the elongated C-terminus and therefore the binding of auranofin. 

## 3. Materials and Methods

### 3.1. Reagents

All reagents used were of the highest available purity. NADP^+^ and NADPH were purchased from Biomol (Hamburg, Germany) and Ni-NTA agarose from Cube Biotech (Monheim am Rhein, Germany). All other reagents were obtained from Merck (Darmstadt, Germany), Roth (Karlsruhe, Germany) or Sigma-Aldrich (Steinheim, Germany). 

### 3.2. Expression and Purification of His-Tagged Ld6PGD wt

For heterologous overexpression in *E. coli*, the gene encoding wt *Ld*6PGD (UniProtKB acc. no. Q18L02) was ordered codon-optimized from Eurofins genomics (Ebersberg, Germany) and subcloned into the pET28a(+) vector, using BamHI and HindIII restriction sites, encoding a N-terminally His-tagged protein. 

*Ld*6PGD wt was heterologously overexpressed in *E. coli* C43 (DE3) cells. pET28a(+) expression plasmid encoding for *Ld*6PGD wt was transformed in *E. coli* C43 (DE3) cells. Cells containing the respective plasmid were grown at 37 °C in lysogeny broth medium (LB) containing 50 µg·mL^−1^ kanamycin. Expression was subsequently induced for 5 h with 0.5 mM IPTG at OD_600_ ~ 0.6. Cells were harvested via centrifugation (15 min, 12,000× *g*, 4 °C), suspended in buffer A (500 mM NaCl, 50 mM Tris, pH 7.8), mixed with protease inhibitors (4 nM cystatin, 150 nM pepstatin, 100 µM PMSF) and stored at −20 °C until cell lysis. For cell lysis and solubilization of the enzyme, lysozyme, DNaseI and 10% glycerol were added to the cell suspension and incubated for ~15 h at 4 °C, followed by three cycles of sonication and centrifugation (30 min, 25,000× *g*, 4 °C). For IMAC purification, the supernatant was applied to a Ni-NTA column, pre-equilibrated with buffer A. Recombinant N-terminally His-tagged proteins were eluted with buffer A containing 50–500 mM imidazole. The fractions containing the recombinant protein were pooled and concentrated using an ultrafiltration unit (Vivaspin 20, 30-kDa filter cut-off; Satorius, Göttingen, Germany). As an additional purification step, and to determine oligomerization state of the respective enzymes, SEC was performed with an ÄKTA FPLC system, using a HiLoad 16/60 Superdex 200 column (GE Healthcare, Freiburg, Germany) pre-equilibrated with buffer A. SEC was performed under native, reductive (5 mM DTT) and oxidative (2 mM H_2_O_2_) conditions as well as in the presence of the substrate (0.2 mM NADP^+^, 0.2 mM 6PG) and product (0.2 mM NADPH). Protein elution was detected at 280 nm and evaluated using the UNICORN software 7.2. The peaks of interest containing the recombinant protein were collected, and concentrated and enzyme purity was assessed with Coomassie-blue-stained SDS-PAGE gels (12% polyacrylamide). The enzyme concentration was determined by measuring the absorbance of the protein solution at 260 and 280 nm using an Eppendorf BioSpectrometer (Hamburg, Germany) and finally calculated using the molecular weight and extinction coefficient of the respective enzymes (*Ld*6PGD wt: 55.4 kDa, 48,610 M^−1^·cm^−1^). After SEC, a final yield of 1–4 mg soluble, pure and active 6PGD per liter *E. coli* culture was achieved for recombinant *Ld*6PGD. The enzyme remained stable for at least ten days when stored at 4 °C and three months when stored at −80 °C with the addition of 250 mM AmSO_4_, as verified via enzyme activity measurements.

### 3.3. Enzymatic Characterization of Ld6PGD wt

6PGD activity was determined at 25 °C by following NADPH [ε_340_ = 6220 M^−1^·cm^−1^] production at 340 nm according to Beutler [75] using the spectrophotometer Evolution 300 (Thermo Scientific, Dreieich, Germany). All measurements were performed in buffer B (50 mM Tris, 3.3 mM MgCl_2_, 0.005% Tween, 1 mg/mL BSA, pH 7.5), whereas the enzyme was pre-diluted in buffer C (50 mM Tris, 0.005% Tween, 1 mg/mL BSA, pH 7.5). The reaction mixture with a final volume of 500 µL contained 200 µM NADP^+^, varying concentrations of the 6PGDs (5–9 nM) and 300 µM 6PG. To determine the apparent *K*_M_ values and V_max_, the substrate (8–500 µM 6PG) and the cosubstrate (8–400 µM NADP^+^) were varied reciprocally. All measurements were performed with three biological independent replicates, each containing at least three measurements, and the apparent kinetic constants were calculated via nonlinear regression using the GraphPad Prism 9.0 software (GraphPad, San Diego, CA, USA). 

### 3.4. Enzymatic Characterization of Gold(I)-Based Compounds 

Dose–response curves of gold(I)-based compounds against *Ld*6PGD were determined in 96-well plates using a Tecan Infinite M200 plate reader (Tecan, Maennedorf, Switzerland). An amount of 10 mM of each compound was dissolved in 100% DMSO and afterwards diluted in assay buffer to the required concentrations (highest final compound concentration of 100 μM). The DMSO tolerance of the enzymes was determined in advance by incubating the enzymes with up to 6% DMSO and determining their activity. Up to this concentration, no impairment of the enzymes’ catalytic activity was observed. In addition, a DMSO control was included in each measurement. 

Different compound concentrations were added to one volume of enzyme mix (final concentrations: 50 mM Tris (pH 7.5), 0.005% Tween 20, 1 mg∙mL^−1^ BSA, 0.7 nM *Ld*6PGD/0.7 nM *Pf*6PGD/0.9 nM *Hs*6PGD) and incubated for ten minutes. To start the reaction, one volume substrate mix (final concentrations: 50 mM Tris (pH 7.5), 0.005% Tween 20, 1 mg∙mL^−1^ BSA, 3.3 mM MgCl_2_, 25/20/25 µM 6PG, 20 µM NADP^+^ for *Ld*6PGD, *Pf*6PGD and *Hs*6PGD) was added to the wells. To increase the sensitivity of the assay, the increase of NADPH was monitored by measuring the fluorescence of NADPH at excitation 340 nm/emission 460 nm (ex340/em460). The substrate mix without 6PG served as a positive control (100% inhibition) and without the compound, but with DMSO as a negative control (100% activity). The assay contained both substrates either in saturation or at concentrations close to *K*_M_ to facilitate the identification of competitive inhibitors. The reaction rate was calculated by dividing the relative fluorescence units (RFU) by time, and only a linear increase in absorbance/fluorescence was considered. IC_50_ values were calculated using GraphPad Prism 9.0 software and Microsoft Excel. 

For the mechanistic characterization of auranofin, either NADP^+^ (0–200 µM) or 6PG (0–250 µM) was titrated at different constant compound concentrations around the IC_50_ (0–30 µM), while keeping the second substrate in saturation (6PG: 300 µM, NADP^+^: 200 µM). To determine the compound mode of inhibition, the relationship between initial velocity (v_0_) and substrate concentrations ([S]) at various constant compound concentrations was analyzed by plotting the data as Michaelis–Menten curves using GraphPad Prism; *K*_M_ values and V_max_ were calculated with GraphPad Prism as well. 

The reversibility of *Ld*6PGD inhibition by auranofin was tested by incubating the enzyme with high compound concentrations, followed by dilution to a compound concentration below IC_50_ and subsequent determination of the enzyme activity. In detail, 14 nM *Ld*6PGD was incubated with 30 µM auranofin for 10 min at room temperature. Afterwards, the enzyme/inhibitor mixture was diluted to a compound concentration below IC_50_ (3 µM) and again incubated for 10 min at room temperature to allow dissociation. Afterwards, NADP^+^ and 6PG were added to start the reaction, either at substrate saturation (300 µM 6PG, 200 µM NADP^+^) or at concentrations close to *K*_M_ (25 µM 6PG, 20 µM NADP^+^). For comparison, controls containing either no inhibitor (100% activity) or inhibitor in the concentration remaining after dilution were prepared. NADPH production was monitored at ex340 nm using the Tecan Infinite M200 plate reader in 96-well plates. Specific activities were calculated from reaction rates. Inhibition was considered irreversible if the activity of the sample after dilution was significantly below the activity of the controls. Inhibition was considered reversible if the activity of the sample was equal to the controls.

### 3.5. Protein Crystallization

For crystallization of *Ld*6PGD wt, the enzyme was concentrated to approximately 8 mg·mL^−1^ in buffer A. A Honeybee 961 crystallization robot (Digilab) was used to prepare the crystallization plates. Drops were generated by mixing 0.2 µL of reservoir solution with 0.2 µL of protein solution. 6PGD crystals were grown in a 96-well format with the sitting-trop vapor diffusion technique at room temperature. Suitable crystals could only be obtained in the presence of the substrate (6PG, 1 mM) and cosubstrate (NADP^+^, 1 mM) in the crystallization buffer (buffer A). The best *Ld*6PGD crystals were obtained in a reservoir containing 24% PEG 3000 and 40 mM of magnesium formate. We produced *Ld*6PGD crystals in complex with NADP(H) diffracting to a resolution up to 3.1 Å. 

### 3.6. Data Collections and Processing

Diffraction data for all crystals were collected at X10SA (detector: DECTRIS EIGER2 XE 16M) of the Swiss Light Source in Villigen, Switzerland, at 100 K and processed with XDS [76]. Before data collection, crystals were soaked in mother liquor with a final concentration of 25% ethylene glycol.

The *Ld*6PGD crystals obey P2_1_2_1_2 space group symmetry and contain two monomers in the asymmetric unit.

### 3.7. Structure Determination and Refinement

The structure was solved by the molecular replacement method. We have generated a search model of *Ld*6PGD by homology modeling with SWISS-MODEL [77] by using the 6PGD dimer from *T. brucei* (PDBID: 1PGJ) as a template, which shares 72% sequence identity with *Ld*6PGD. The first refinement revealed an initial R_free_ of 40%, comprising residues 15–478 of both monomers. After several rounds of model correction and completion of the model with the interactive graphics program Coot, the final R_free_ value dropped to 31.7%. The structure was refined to a resolution of 3.1 Å. The asymmetric unit contains two PGD monomers (A, B), two NADP(H), two phosphate ions and one water molecule. The two ions and one NADP(H) molecule could be clearly determined in the first difference map, the second NADPH was only partially visible. The overall temperature factor of the structure is relatively high at 125 Å^2^ but fits well with the measured Wilson B factor of 129 Å^2^. Despite the moderate resolution, the electron density of the structure is well defined for the entire model (1–478), indicating good phases due to a matching model. During refinement, 10% of all reflections were omitted and used for calculation of an R_free_ value, the final statistics are shown in Table 2. 

The PHENIX program suite [78,79] served for reflection phasing and structure refinement. The interactive graphics program Coot [80] was used for model building. The structures were superimposed using the SSM algorithm tool [81], implemented in the Coot graphics package. The SSM tool is a structural alignment based on secondary structure matching. Molecular graphics images were produced using the UCSF Chimera package [82]. 

## 4. Conclusions

Since resistance against commonly used antileishmanial drugs such as miltefosine [25], amphotericin-B [26] and sodium antimony gluconate [27] has already emerged [3,4,5], new drug targets and antileishmanial agents are urgently required. The crystal structure and biochemical characterization of *Ld*6PGD presented in this study provides a first step to validate 6PGDs from *Leishmania* parasites as a promising antitrypanosomatid drug target. In addition, we suggest gold(I)-containing compounds as a promising class of selective *Ld*6PGD inhibitors. As the human 6PGD was not inhibited at micromolar concentrations, we postulate that the elongated C-terminal domain of the human enzyme sterically prevents auranofin from binding to the 6PG active site, demonstrating that selective inhibition of this conserved enzyme is possible. Co-crystallization studies are underway to investigate the exact *Ld*6PGD binding site of auranofin, or rather of the gold moiety, and will provide the basis for further structure-based activity relationship studies. 

## Figures and Tables

**Figure 1 ijms-24-08615-f001:**
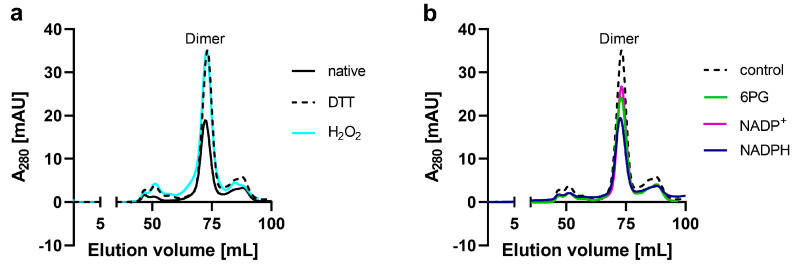
SEC analysis of recombinant *Ld*6PGD. Full-length *Ld*6PGD wt was gel filtrated on a HiLoad 16/60 Superdex 200 column pre-equilibrated in buffer A (500 mM NaCl, 50 mM Tris, pH 7.8). (**a**) SEC profiles of *Ld*6PGD wt under native, reductive and oxidative conditions. SEC of *Ld*6PGD wt under native conditions (black, rv = 72.3 mL ≙ 93.9 kDa), in the presence of 5 mM DTT (black-dotted, rv = 73.2 mL ≙ 87.3 kDa) and 2 mM H_2_O_2_ (cyan, rv = 72.8 mL ≙ 89.9 kDa) revealed identical elution patterns with one peak equivalent to a dimer. (**b**) SEC profiles of *Ld*6PGD wt in the presence or absence of ligands. SEC in the presence of 0.2 mM 6PG (green, rv = 72.95 mL ≙ 88.8 kDa), in the presence of 0.2 mM NADP^+^ (magenta, rv = 73.2 mL ≙ 87.2 kDa) and 0.2 mM NADPH (purple, rv = 72.6 mL ≙ 91.5 kDa). For comparative reasons, a control without ligands is shown (black-dotted, rv = 73.2 mL ≙ 87.3 kDa). Representative chromatograms (*n* ≥ 2) are shown for each condition.

**Figure 2 ijms-24-08615-f002:**
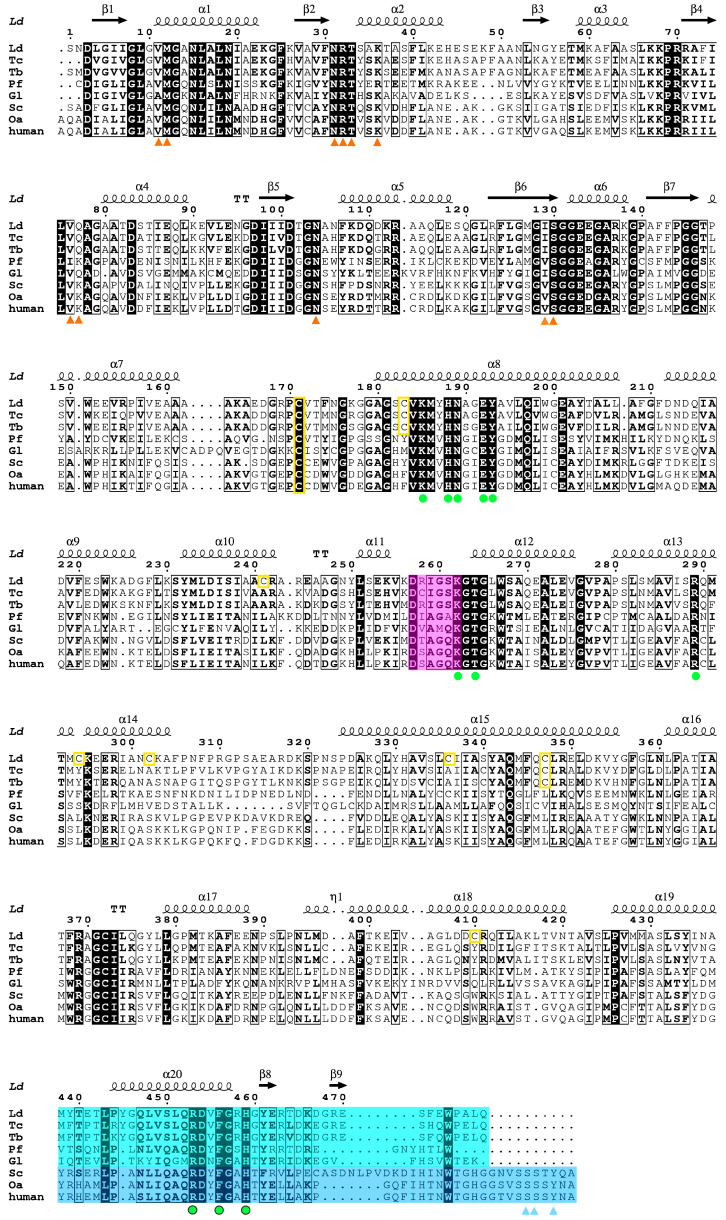
Structure-based multiple sequence alignment of 6PGDs from various species. Shown 6PGD sequences correspond to *Leishmania donovani* (Ld; acc. no. Q18L02), *Trypanosoma cruzi* (Tc; acc. no. Q6WAT5), *Trypanosoma brucei brucei* (Tb; acc. no. P31072), *Plasmodium falciparum* (Pf; acc. no. Q8IKT2), *Giardia lamblia* (Gl; acc. no. A8BWM8), *Saccharomyces cerevisiae* (Sc; acc. no. P38720), *Ovis aries* (Oa; acc. no. P00349) and *Homo sapiens* (human; acc. no. P52209). Strictly conserved residues are highlighted with black background, while highly similar residues, with similar physicochemical properties in at least six out of the eight sequences, are shown in bold letters. The secondary structure elements of *Ld*6PGD (PDBID: 8C79) are displayed above the sequence alignment. Residues of the NADP^+^ binding site are displayed as orange triangles. Blue triangles correspond to the *Hs*6PGD sequence and indicate additional residues, which are involved in NADP^+^ binding. The 6PG binding site comprises residues from both subunits and is therefore displayed as green dots for one subunit and green dots with black outlines for the adjacent subunit. Yellow-colored boxes indicate cysteines of the *Ld*6PGD structure. Residues of the active site loop (D257-T262) are highlighted in magenta. Residues of the C-terminal domain of protozoan species and multicellular species with elongated C-terminus are highlighted in cyan and blue, respectively. The structure-based multiple sequence alignment was performed using the program ESPript 3.0.

**Figure 3 ijms-24-08615-f003:**
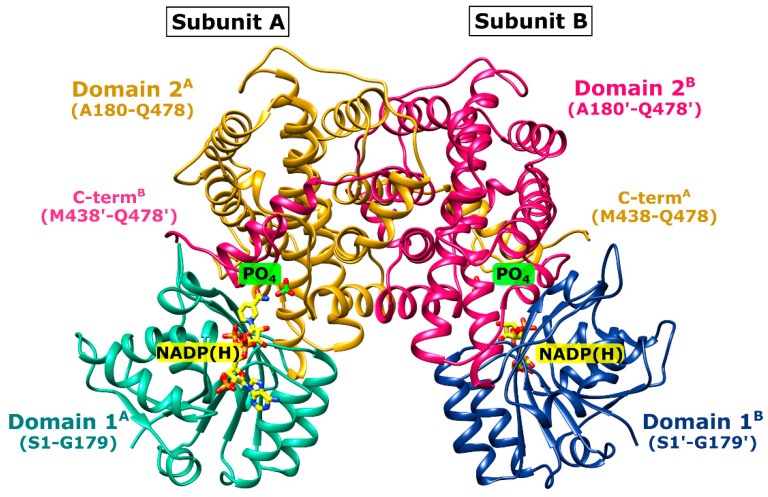
Overview of the *Ld*6PGD wt dimer. *Ld*6PGD wt dimer in complex with NADP(H) and phosphate (PO_4_) is shown in ribbon presentation. The “Rossmann-like” domain 1 (S1-K179) of subunits A (domain 1^A^) and B (domain 1^B^) is colored light green and navy blue, respectively. Domain 2 (A180-Q478) of subunits A (domain 2^A^) and B (domain 2^B^) is colored gold and magenta, respectively. The C-terminal domain of domain 2 comprises residues M348-Q478. Within the 6PG binding site, a PO_4_ ion (green) is visible instead of the substrate 6PG. The cosubstrate NADP(H) (yellow) and the PO_4_ (green) of subunit A and B are shown as stick models.

**Figure 4 ijms-24-08615-f004:**
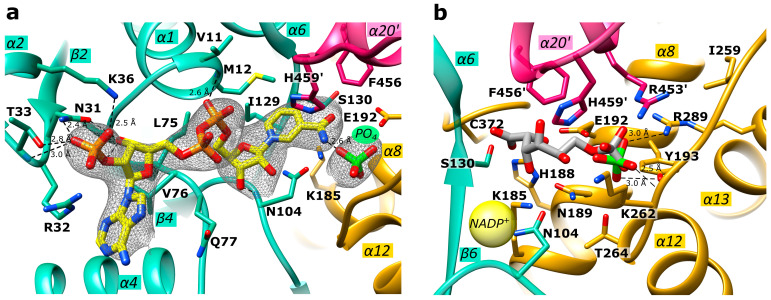
Active site close-up of *Ld*6PGD. The active site of monomer A is shown. Ribbons and residues of domain 1 are colored light green and of domain 2, gold. The C-terminal part of domain 2 of monomer B is colored magenta. (**a**) Close-up of the NADP(H) binding pocket within the “Rossmann-like” domain. The NADP(H) moiety is colored yellow. Within the 6PG binding pocket, a phosphate ion (PO_4_) is visible (green). Electron density map (F_O_-F_C_ polder map) contoured at 3.0 σ for NADP(H) and PO_4_ is shown in black. (**b**) Close-up of the 6PG binding pocket. 6PG (gray) is shown from the superimposed *Pf*6PGD structure (PDBID: 6FQZ) with the PO_4_ moiety at the same position as the PO_4_ visible in our structure (green). The approximate position of NADP^+^ is marked with a yellow ball. Residues of the NADP^+^ and the 6PG binding pocket are shown in stick models, and hydrogen bonds are indicated with black dotted lines.

**Figure 5 ijms-24-08615-f005:**
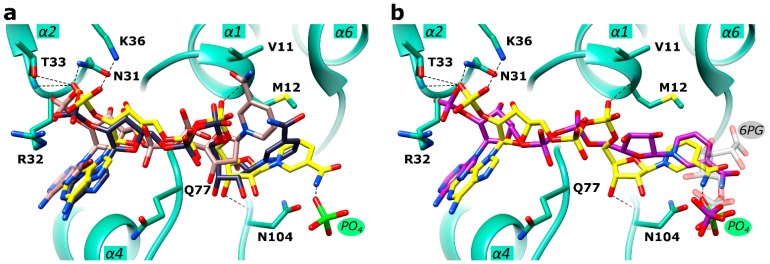
Comparison of NADP^+^ and NADPH conformations in *Ld*6PGD and orthologues. The structures from *Ld*6PGD, human (*Hs*), *Ovis aries (Oa)* and *Plasmodium falciparum* (*Pf*) are superimposed, but for clarity, only the ribbon and residues of domain 1 of the *Ld*6PGD structure are shown and colored light green. (**a**) NADP(H) from the *Ld*6PGD structure is colored yellow. The NADP^+^ molecule from the NADP^+^-complexed structures of *Hs*6PGD (PDBID: 2JKV) and of *Oa*6PGD (PDBID: 1PGN) is colored navy blue and nude, respectively. (**b**) NADP(H) from the *Ld*6PGD structure is shown in yellow and NADPH from the NADPH-complexed *Oa*6PGD structure (PDBID: 1PGO) in purple. 6PG (transparent) is taken from the *Pf*6PGD structure (PDBID: 6FQZ) with the phosphate moiety at the same position as the PO_4_ (green) visible in our structure. In this conformation, 6PG would clash with bound NADP(H) of the *Ld*6PGD and the NADPH-complexed *Oa*6PGD structure. Residues of the NADP^+^ and the 6PG binding pocket are shown in stick models, and hydrogen bonds are indicated with black dotted lines.

**Figure 6 ijms-24-08615-f006:**
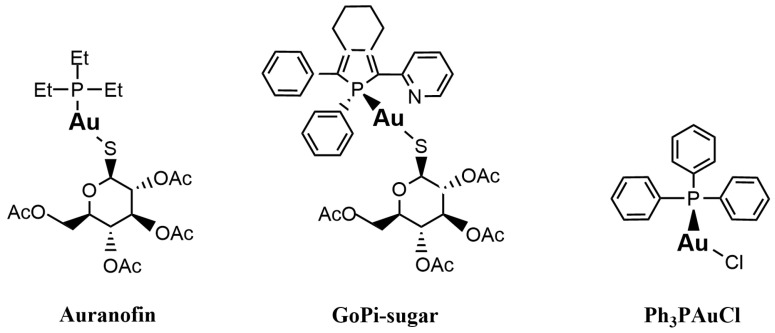
Chemical structures of the gold(I)-based compounds auranofin, GoPi-sugar and Ph_3_PAuCl (chloro(triphenylphosphine)gold(I)).

**Figure 7 ijms-24-08615-f007:**
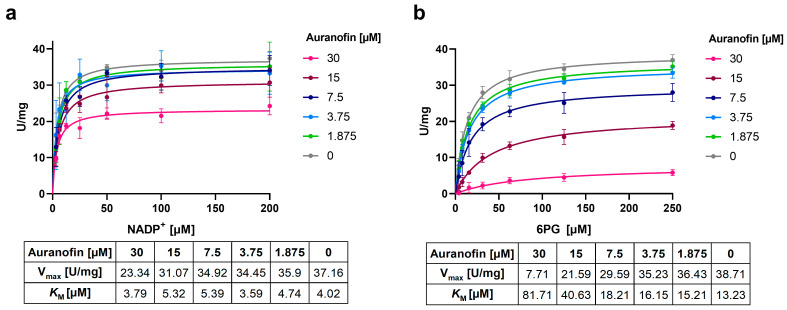
Mechanism of inhibition of auranofin against *Ld*6PGD. Various compound concentrations were titrated against (**a**) NADP^+^ or (**b**) 6PG. Michaelis–Menten graphs of [U/mg] against NADP^+^ [µM] and 6PG [µM], respectively, are shown. (**a**) The *K*_M_ value for NADP^+^ stays constant with a minor reduction in V_max_, indicating a non-competitive inhibition of auranofin against NADP^+^. (**b**) The *K*_M_ value for 6PG increases and V_max_ decreases with increasing compound concentrations, indicating a mixed type of inhibition. Representative graphs from two independent replications are shown, each including at least two measurements.

**Figure 8 ijms-24-08615-f008:**
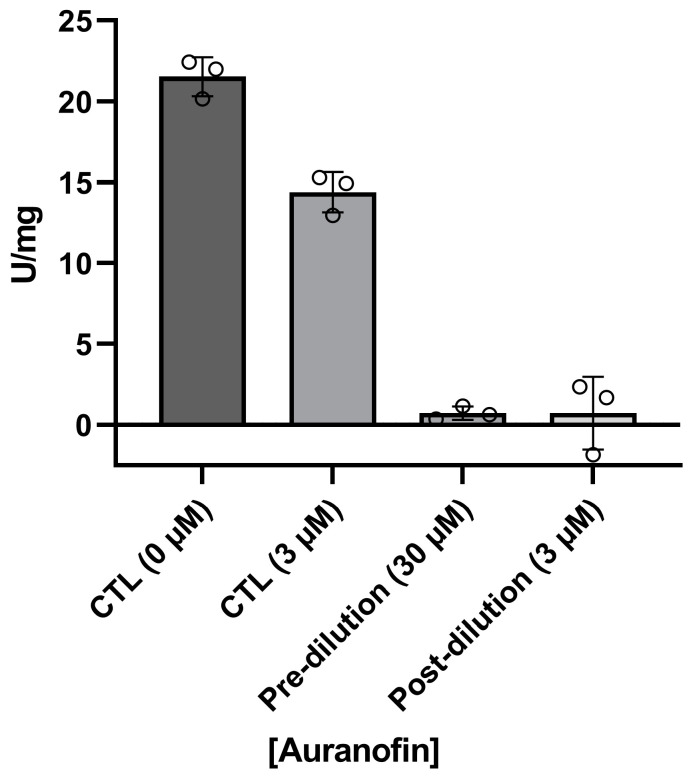
Reversibility of *Ld*6PGD inhibition by auranofin. Reversibility of *Ld*6PGD inhibition by auranofin was determined by incubating the enzyme with a high compound concentration of 30 µM (pre-dilution), followed by dilution to 3 µM (post-dilution). The undiluted sample was completely inhibited, and dilution did not restore activity compared to the control containing the same final compound concentration (CTL, 3 µM). Therefore, the inhibition of *Ld*6PGD by auranofin is irreversible. Treatment without auranofin was used as 100% activity control (CTL, 0 µM). The measurements were performed using the substrates 6PG and NADP^+^ at *K*_M_ concentrations. Values are expressed as mean ± SD from three independent measurements with different enzyme batches, each including three measurements.

**Figure 9 ijms-24-08615-f009:**
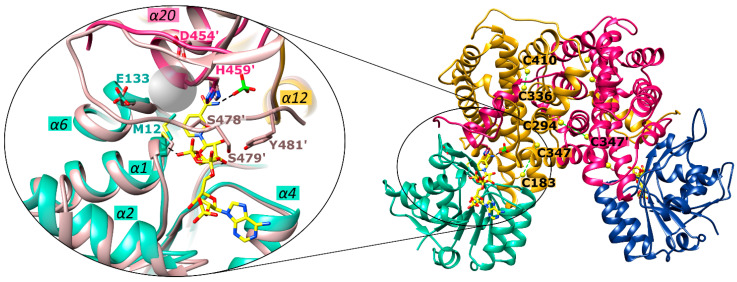
Potential binding site of gold(I) complexes in *Ld*6PGD. The structures from *Ld*6PGD and *Hs*6PGD (PDBID: 2JKV) are superimposed and residues from the *Ld*6PGD structure that are potentially involved in binding of the gold moiety from auranofin and other gold-containing compounds are shown. Ribbons and residues of domain 1^A^ and 1^B^ from the *Ld*6PGD structure are colored light green and navy blue, and of domain 2^A^ and domain 2^B^, gold and magenta, respectively. NADP(H) and the phosphate moiety from the *Ld*6PGD structure are colored yellow and green, respectively. Ribbons and residues from the superimposed *Hs*6PGD structure are colored nude, and additional residues from the elongated C-terminus of the *Hs*6PGD structure which are involved in NADP^+^ binding are shown. In addition, cysteines that are absent in the human structure are shown in the overall view and potential cysteine pairs linked by a green dotted line.

**Table 1 ijms-24-08615-t001:** Comparative kinetic parameters of recombinant 6PGD from *Leishmania donovani* and different homologous species.

	6PG				NADP^+^			
	V_max_ (U·mg^−1^)	*K*_M_ (µM)	*k*_cat_ (s^−1^)	*k*_cat_/*K*_M_ (µM^−1^·s^−1^)	V_max_ (U·mg^−1^)	*K*_M_ (µM)	*k*_cat_ (s^−1^)	*k*_cat_/*K*_M_ (µM^−1^·s^−1^)
*Leishmania* *donovani*	33.3 ± 5.1	20.2 ± 2.9	30.8 ± 4.7	1.5 ± 0.02	33.1 ± 5.2	13.7 ± 1.2	30.5 ± 4.8	2.2 ± 0.2
*Leishmania* *donovani* [47]	n/a	80.2 ± 7.4	1.0 ± 0.02	0.01	n/a	22.4 ± 2.7	0.9 ± 0.02	0.04
*Trypanosoma cruzi* [60]	32	22.2	n/a	n/a	32	5.9	n/a	n/a
*Trypanosoma brucei* [59,60]	31.3	3.5	27	7.7	31.2	1.5	27	18
*Plasmodium* *falciparum* [49]	8.0 ± 1.8	11.3 ± 2.7	7.1 ± 2.5	0.6	8.0 ± 1.8	9.0 ± 4.2	7.6 ± 2.2	0.84
*Giardia* *lamblia* [58]	26.2	49.2	n/a	n/a	26.2	139.9	n/a	n/a
Human [49]	22.1 ± 1.2	33.7 ± 7.1	22.2 ± 0.3	0.6	22.1 ± 1.2	6.9 ± 2.0	21.4 ± 0.8	3.1
Sheep liver [57]	18.8 ± 0.9	16.1 ± 1.3	n/a	n/a	18.8 ± 0.9	6.76 ± 1.6	n/a	n/a

n/a—not available. Values are expressed as mean ± SD from at least three independent determinations with different enzyme batches, each including at least three measurements.

**Table 2 ijms-24-08615-t002:** Data collection and refinement statistics.

PDB Code	8C79
**Data collection**	
Space group	P 21 21 21
Cell dimensions	
a, b, c (Å)	65.87, 117.35, 128.91
α, β, γ (°)	90, 90, 90
Resolution (Å)	46.1–3.1 (3.2–3.1)
R-merge (%)	8.7 (184.6)
*I/σI*	9.9 (0.9)
Completeness (%)	99.57 (99.95)
Redundancy	5.1 (5.2)	
Molecules per AU	2
Wilson B-factor	129.0
CC_1/2_ (%)	99.8 (34.8)
**Refinement**	
Resolution (Å)	3.1
No. reflections	18,687
*R_work_*/*R_free_* (%)	26.2 (41.3)/31.7 (45.0)
No. atoms	
Proteins	7278
Ligands	87	
Water	1
Protein residues	956
B-factors	
Proteins	126.6
Ligands	121.0
Water	93.8
Ramachandran plot (%)	
Favored (%)	97.5
R. m. s. deviations	
Bonds lengths	0.011
Bond angles	0.94

Values in parentheses are for the highest-resolution shell.

**Table 3 ijms-24-08615-t003:** Biochemical activities of auranofin, GoPi-sugar and Ph_3_PAuCl.

	IC_50_ [µM] for Recombinant Protein		
Compound	*Ld6PGD*	*Pf6PGD*	*Hs6PGD*
Auranofin	8.6 ± 1.0	20.1 ± 1.5	>100
GoPi-sugar	9.4 ± 2.1	n.d.	n.d.
Ph_3_PAuCl	2.1 ± 0.3	n.d.	n.d.

n.d.—not determined. Values are expressed as mean ± SD from three independent determinations with different enzyme batches, each including three measurements.

## Data Availability

Coordinates and measured reflection amplitudes have been deposited in the Worldwide Protein Data Bank RCSB PDB (http://pdb.org): code 8C79 for *Ld*6PD wild type in complex with NADP(H).

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
