# Peer review of "Structure of Leishmania donovani 6-Phosphogluconate Dehydrogenase and Inhibition by Phosphine Gold(I) Complexes: A Potential Approach to Leishmaniasis Treatment"

_ijms, 2023, doi:10.3390/ijms24108615_

Round 1
Reviewer 1 Report
The authors solved the structure of G6PD from Leishmania donovani and biochemically characterized the enzyme. Leishmania donovani causes black fever in humans and thus this study is important for human health. Overall, the article is well-written, and the results are well-presented. However, I have a few minor concerns.
1- The value of Rwork and Rfree are very high for the resolution and the difference between Rwork and Rfree is also very high.
2- There are several residues with non-rotameric side chains. Does the density justify the particular rotamer? If not, then why most preferred rotamers were not selected during the model building?
3- NADP has several bond lengths and bond angle outliers. Were attempts made to fix the issues?
4- From lines 179-182, the authors state that there is a clear density of one NADPH molecule in each subunit. At the same time, it is stated that the clear density for nicotinamide and adenine moiety is missing. These are contradictory statements.
5- Can the author model the nicotinamide and adenine moiety in chain B as well as setting its occupancy to 0?
6- In Figure 4, the polder map is shown around NADP, which is fine but omit map should also be shown at least in the supplement.
7- In Figure 4, the hydrogen bonding interactions should be shown with distance.
